# Exploring functional microbiota for uranium sequestration in Zoige uranium mine soil

Xiang Wang,[1] Li Zhao,[2] Xu Zhang,[3] Yanxia Wei,[1] Aixia Lu,[1] Jian Zhou,[1,4] Guiqiang He[1,4]

**ABSTRACT** The Zoige uranium mine is situated in the harsh, cold northern region of Sichuan, characterized by its high altitude and fragile ecosystem. Uncovering the organisms that thrive in such extreme climates, particularly microorganisms, is of paramount importance for advancing bioremediation efforts. Herein, the potential functional microbiota for uranium sequestration in Zoige uranium mine soil was explored using high-throughput sequencing combined with bioinformatics analysis. Analysis of the physicochemical properties of soils showed that the concentration of uranium ranged from 35.20 to 40.62 $\mu g \cdot g^{-1}$ around the uranium mine. Bacterial communities differed significantly in soils around the Zoige uranium mine, with the most abundant phyla being Actinobacteria, Proteobacteria, Acidobacteria, Chloroflexi, Gemmatimonadota, Verrucomicrobia, and Firmicutes. Notably, Actinobacteria was considered a biomarker for distinguishing soils with high uranium by linear discriminant analysis effect size. Meanwhile, the correlation analysis demonstrated that Firmicutes and Cyanobacteria were significantly and positively associated with uranium in soil samples, with the correlation coefficients being 0.8601 and 0.7832, respectively. Furthermore, the phylogenetic investigation of communities by reconstruction of unobserved states analysis revealed that the bacterial microbiota was mainly enriched in biosynthesis function in these soils. Interestingly, the abundance of functional genes involved in amino acid biosynthesis increased whereas that related to fatty acid biosynthesis decreased with an increase in uranium content. Taken together, Actinobacteria, Firmicutes, and Cyanobacteria were the potential functional microbiota for uranium sequestration via amino acid and fatty acid biosynthesis pathways in Zoige uranium mine soil. These findings are conducive to obtaining functional strains for developing microbial remediation technologies for uranium contamination.

**IMPORTANCE** Based on the significance of the Zoige uranium mine and its unique ecological environment, this study emphasizes the necessity of *in situ* bioremediation. Herein, the potential functional microbiota for uranium sequestration in Zoige uranium mine soil was explored using high-throughput sequencing and bioinformatics analysis. Actinobacteria, Firmicutes, and Cyanobacteria were the potential functional microbiota in Zoige uranium mine soils. These microbes interacted and tolerated uranium via amino acid and fatty acid biosynthesis pathways. These findings provide insights into the functional microbiota of uranium sequestration, which are conducive to developing microbial resources and bioremediation technology for treating uranium contamination.

**KEYWORDS** uranium mine, soil community, uranium sequestration, microbial remediation

**Peer Reviewer** Xugang Dang, Shaanxi University of Science and Technology, Xi'an, China

Address correspondence to Guiqiang He, guiqianghe@swust.edu.cn, or Jian Zhou, zhoujian@swust.edu.cn.

The authors declare no conflict of interest.

See the funding table on p. 11.

Natural uranium, a critical strategic resource and essential energy mineral, is widely used in nuclear power generation, medical treatment, and various industrial processes (1, 2). With the prospect of nuclear energy becoming more and more bright,

the production of uranium mining is therefore on the increase year by year. Currently, over 70% of the total amount of high-quality uranium resources are found in uranium deposits of China (3, 4). Among them, the Zoige uranium mine in Sichuan province is a crucial carbonaceous-siliceous-argillitic rock-type uranium deposit in China. It has been instrumental for China to achieve basic self-sufficiency in uranium resources (5).

Unfortunately, uranium mining inevitably contaminates the surrounding soil, causing groundwater, vegetation, and even ecosystem pollution (6–8). In recent years, researchers have conducted extensive studies on technologies for remediating uranium-contaminated soil by physical and chemical methods. For example, batch synthesis of phosphate-functionalized magnetic calcium alginate hydrogel by an *in situ* co-precipitation method exhibits excellent uranium enrichment capabilities in uranium-contaminated soil (9). However, physical and chemical methods possess inherent limitations, including exorbitant costs, the risk of disrupting the structural and physicochemical characteristics of soil, and the induction of secondary pollution.

Many researchers, therefore, did much exploration on microbial community of uranium mine soils for developing the bioremediation technology. Furthermore, studies have shown that there are significant differences in microbial community between different uranium mine soils because of its peculiar geological environment and climatic conditions (10, 11). For example, Firmicutes and Betaproteobacteria were the most dominant bacterial phyla in a uranium mine of northern Saskatchewan, while the uranium tailing of Bulgaria harbored the dominant microbial taxa, mainly including Proteobacteria, Acidobacteria, and Bacteroidetes (11, 12). Zoige uranium mine is a hard-rock type base in China, with the largest reserves of uranium resource. This uranium deposit is located in a high-altitude area with a fragile ecological environment, so revealing the microbial community in this extreme environment is meaningful for indigenous ecological remediation.

Delightedly, some functional strains have been isolated and applied for bioremediation of uranium pollution. For instance, *Bacillus*, *Actinomycetes*, and some phosphorus-solubilizing strains could effectively immobilize the uranium in soils by bioaugmentation, consequently preventing migration of soluble uranium (13–15). It is worth noting that *in situ* bioremediation with indigenous functional strains can significantly improve the remediation efficiency (16, 17). Therefore, in terms of *in situ* bioremediation, it is vital to unveil the microbial community and identify the functional microbiota in the contaminated areas.

In the current study, soil samples were collected from the Zoige uranium mine, with sampling locations strategically positioned to the east, south, west, and north of the uranium deposit's central point (Fig. S1). The uranium content and bacterial community in these soils as well as their correlations were comprehensively analyzed. Furthermore, biomarker and functional prediction were revealed to gain deeper insights into the functional microbiota for uranium sequestration by linear discriminant analysis effect size (LEfSe) and phylogenetic investigation of communities by reconstruction of unobserved states (PICRUSt2), respectively. This study aims to identify the functional microbiota for uranium sequestration in Zoige uranium mine, which may help develop *in situ* remediation technologies for uranium contamination using specific functional strains.

## MATERIALS AND METHODS

### Soil samples source and collection

Soil samples were collected from the Zoige uranium deposit, which is located in the ecologically fragile Northwest Sichuan Plateau of China. It is categorized as one of the four primary types of uranium deposits found in China. The area boasts a temperate-to-subtropical monsoon climate, characterized by an average annual temperature ranging from 5.6°C to 8.9°C and an altitude of approximately 3,500 to 4,000 m (18).

Sampling points were located 100 m to the east (E), west (W), south (S), and north (N) of the uranium deposit's central point. Parallel samples were collected along a straight line, spaced at intervals of 10 m. At each designated point, surface materials were removed prior to conducting random sampling. Samples collected in the same direction were mixed, yielding a total weight of 500 g per sample. In total, 12 soil samples (arranged as 4 groups of 3) were gathered at depths of 0, 10, and 20 cm, with each sample having a thickness ranging from 2 to 3 cm. The sampling site locations were depicted in Fig. S1. The soil samples were temporarily stored in a refrigerator at 4°C. One portion of these samples was designated for bacterial community analysis, and the other portion was used to assess their physical and chemical properties.

## Determination of soil physicochemical properties

The pH of the soil was determined using a PHS-3CW pH meter (Shanghai Bante Instrument Inc., Shanghai, China) (19). The moisture content (MC) of soils was determined using the constant gravimetric method (20). Total phosphorus (TP) and uranium content (UC) were determined after soil digestion. TP and available phosphorus (AP) of soil were determined using the ammonium-molybdate method (21). Ammonium nitrogen (AN) content was determined by the indophenol blue colorimetric method (22). The electrical conductivity (EC) of soil was determined using a conductivity meter (ratio of soil to water, 1:5 dry [wt/vol]) (Shanghai Bante Instrument Inc., Shanghai, China). Uranium was determined by Arsenazo III spectrophotometry (23).

## DNA extraction, PCR amplification, and amplicon detection

Total DNA was extracted separately from the soil (4 × 3) using the OMEGA Soil DNA Kit (M5636-02) (Omega Bio-Tek, Norcross, GA, USA) and quantified using a NanoDrop spectrophotometer. DNA quality was detected by 1.2% (wt/vol) agarose gel electrophoresis. The V3-V4 region of the 16S ribosomal RNA (rRNA) gene was amplified using a universal primer pair 338F (ACTCCTACGGGAGGCAGCA) and 806R (GGAC-TACHVGGGTWTCTAAT) (24). Purified PCR products were measured using a microplate reader (BioTek, FLx800), and sequencing library fragments were prepared. The sample library was diluted and mixed in equal amounts as pooled samples. Qualified samples were sent to Shanghai Personal Biotechnology Co., Ltd. (Shanghai, China) for high-throughput sequencing.

## High-throughput sequencing

Paired-end sequencing of community DNA fragments was performed using the Illumina MiSeq/NovaSeq platform. Sequencing data were analyzed using the Quantitative Insights Into Microbial Ecology (QIIME) pipeline (25). Primer removal, mass filtering, denoising, splicing, and chimera removal were performed using the Divisive Amplicon Denoising Algorithm 2 (DADA2) method (26). Nucleic acid sequences were corrected using FrameBot software (v1.2), and the sequence length distributions were statistically performed using R scripts. Bacterial species were annotated using the Greengenes database (Release 13.8, https://ngdc.cncb.ac.cn/databasecommons/database/id/3120) (27). Mafft was applied for amplicon sequence variant (ASV) alignment, and FastTree was applied for phylogeny construction (28, 29). The abundance table of ASVs was leveled using the QIIME feature-table rarefy function. The leveling depth was set to 95% of the minimum sample sequence size (30).

## Bioinformatics analysis

The significance of the alpha diversity difference was verified using the Kruskal-Wallis rank-sum test and the Dunn test. A Venn diagram was constructed using R software to identify common ASVs in all samples. The taxonomic status of the shared ASVs was determined using the annotated information of the Ribosomal Database Project database. Principal coordinates analysis (PCoA) was used to evaluate heterogeneity in

**TABLE 1** Physicochemical properties of soil samples (mean ± standard error)[a,b]

| Physicochemical properties | East | North | South | West |
|---|---|---|---|---|
| pH | 6.71 ± 0.22a | 6.69 ± 0.18a | 6.67 ± 0.06a | 6.90 ± 0.01a |
| EC (µs/cm) | 145.6 ± 14.77ab | 126.75 ± 7.83b | 165.1 ± 19.74a | 44.05 ± 3.40c |
| MC (%) | 22.26 ± 0.31a | 15.60 ± 0.11b | 4.41 ± 0.18d | 12.71 ± 0.15c |
| TP (mg/100 mg) | 16.88 ± 0.02a | 14.50 ± 0.01b | 10.75 ± 0.01c | 10.25 ± 0.04d |
| AP (mg/100 mg) | 0.37 ± 0.04a | 0.16 ± 0.01b | 0.16 ± 0.01b | 0.16 ± 0.02b |
| AN (mg/100 mg) | 5.07 ± 0.06a | 4.24 ± 0.01b | 4.03 ± 0.02d | 4.15 ± 0.01c |
| UC (µg/g) | 35.20 ± 0.08d | 40.31 ± 0.02b | 36.73 ± 0.03c | 40.62 ± 0.08a |

[a]AN, ammonium nitrogen.
[b]Different lowercase letters within the same row represent significant differences.

species composition utilizing Bray-Curtis distances. LEfSe analysis was performed to identify a stable biomarker between the groupings using linear discriminant analysis (LDA) score thresholds of >4. The correlation analysis was performed using a genescloud platform (https://www.genescloud.cn). The 16S rRNA gene sequence was predicted using PICRUSt2 software, based on Kyoto Encyclopedia of Genes and Genomes (KEGG, https://www.kegg.jp/) functional databases.

## Statistical analysis

Statistical analysis was performed using SPSS software (version 29; IBM SPSS Inc., NY, USA) and R software (version 4.2.3; R Foundation for Statistical Computing, Vienna, Austria). The *t*-test was employed to compare differences between two groups of samples, while Tukey's test was used for multiple comparisons. All experiments were conducted in triplicate. *P* < 0.05 was considered statistically significant.

## RESULTS

### Determination of soil physicochemical properties

Differences in the physicochemical properties and uranium contents of soil samples from the Zoige uranium ore field were shown in Table 1. The soil pH ranged from 6.67 to 6.90, suggesting that Zoige uranium soil was slightly acidic. The soil MC ranged from 4.41% to 22.26%, with samples east and south exhibiting the highest and the lowest MC, respectively. The AN ranged from 4.15 to 5.07 mg·100 mg$^{-1}$. The TP ranged from 10.2 to 16.88 mg·100 mg$^{-1}$, with significant differences in soils among the four regions (*P* < 0.05). The soil AP content ranged from 0.16 to 0.37 mg·100 mg$^{-1}$, with an average value of 0.22 mg·100 mg$^{-1}$ and significant differences in soils among the four regions (*P* < 0.05). UC differed significantly in soils among the four regions (*P* < 0.05). The order of concentrations of sample uranium was west > north > south > east soils. Sample group west had the highest UC (40.62 µg·g$^{-1}$), whereas sample group east had the lowest UC (35.20 µg·g$^{-1}$).

### Soil microbial diversity

Four soil samples were subjected to high-throughput sequencing. The resulting fragment length ranged from 218 to 442 bp, with an average length of 383 bp (Fig S2a). There were 1,624,275 raw sequences before quality control (QC) and demosaicing. DADA2 QC and demosaicing yielded 1,037,266 usable sequences. Based on the principle that the minimum number of effective sequences is slightly greater than the number of draws, an equal amount of leveling was carried out, yielding 35,563 draw-and-level sequences. The Chao1 index sparse curve tended to reach saturation plateau, indicating that the sequence coverage was sufficient to obtain the bacterial community diversity (Fig. S2b). The Venn diagram was utilized to determine the specificity and similarity of soil samples around the uranium mine at the ASV level. Sample groups west, east, north, and south had 7,678, 6,542, 6,110, and 6,997 ASVs, respectively. A total of 1,295 samples were shared among the 4 groups, accounting for 16.87%–21.19% (Fig. 1a).

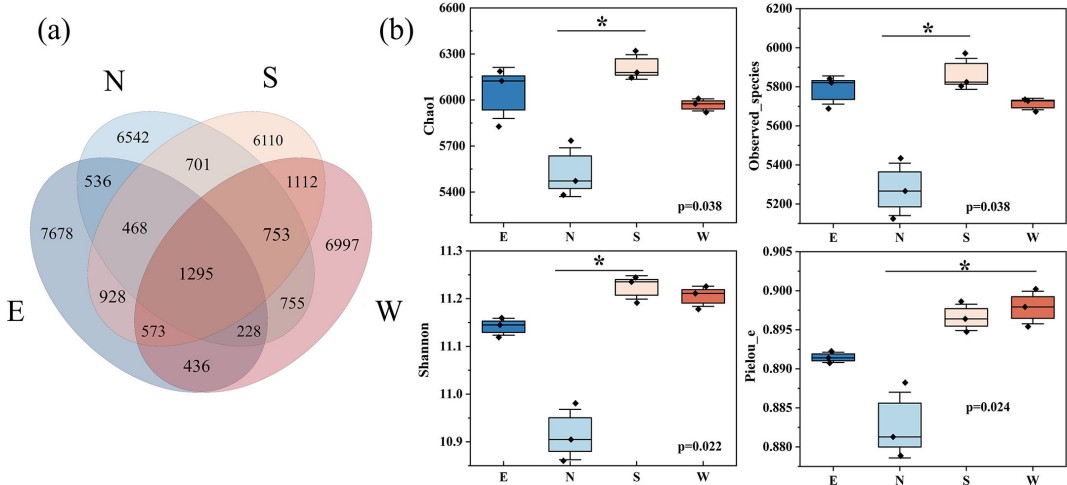

**FIG 1** Comparisons of operational taxonomic units (ASVs) by Venn analysis (a) and α-diversity indices among soil samples (b). The asterisk indicates significant differences ($P < 0.05$) analyzed by the Kruskal-Wallis test.

α-Diversity is used to characterize the diversity of the bacterial community in a sample. Chao1, Shannon, and Pielou's evenness, respectively, were used to characterize richness, diversity, and uniformity. Results showed that these α-diversity indices were lower in the north soils than those in the other three samples (Fig. 1b). North soils had significantly lower community diversity ($P < 0.05$), richness ($P < 0.05$), and evenness ($P < 0.05$) than those of south soils, suggesting that richness and diversity of bacterial communities were lower in the natural wetland. Significant differences in bacterial diversity were observed between the four soil samples, and the north soils had the lowest soil bacterial richness.

## Soil microbial community structure

In total, 27 phyla, 77 classes, 157 orders, 235 families, and 356 genera of bacteria were obtained in all samples (Fig. 2a). No significant difference was found in the soil bacterial community structure at each level of the four sampling sites and the four sample groups (Fig. 2b). The relative abundance of the top 10 species at the phylum and class levels is depicted in Fig. 2c and d. The top 10 most abundant phyla were Actinobacteria, Proteobacteria, Acidobacteria, Chloroflexi, Gemmatimonadetes, Verrucomicrobia, Rokubacteria, Bacteroidetes, Nitrospirae, and Firmicutes. Actinobacteria was the most predominant phylum, with an abundance range of 26.11%–36.49%, followed by Proteobacteria, with an abundance range of 19.5%–30% (Fig. 2c). The top 10 most abundant bacterial classes were Thermoleophilia, Alphaproteobacteria, Actinobacteria, Gammaproteobacteria, Subgroup_6, Blastocatellia, Gemmatimonadetes, KD4-96, Verrucomicrobia, and Acidimicrobiia. Thermoleophilia (10.99%–15.41%), Alphaproteobacteria (9.49%–12.22%), and Actinobacteria (7.66%–12.22%) were the three most dominant classes (Fig. 2d).

## Species differences and marker species analysis

β-Diversity was described in terms of PCoA-weighted unit distances. Samples with a similar community structure tended to cluster together. In the PCoA plot, PCo1 and PCo2 explained 33.8% and 21.8% of the variance, respectively (Fig. 3a). The non-overlapping ASV distributions and dissimilar community structures observed among the four sampling points suggest the presence of distinct microbial communities.

Cluster analysis of four samples was carried out using ORIGIN software to explore the relationship between the bacterial community structure of soil samples in different regions. The clustering results were integrated with the relative abundance of species at

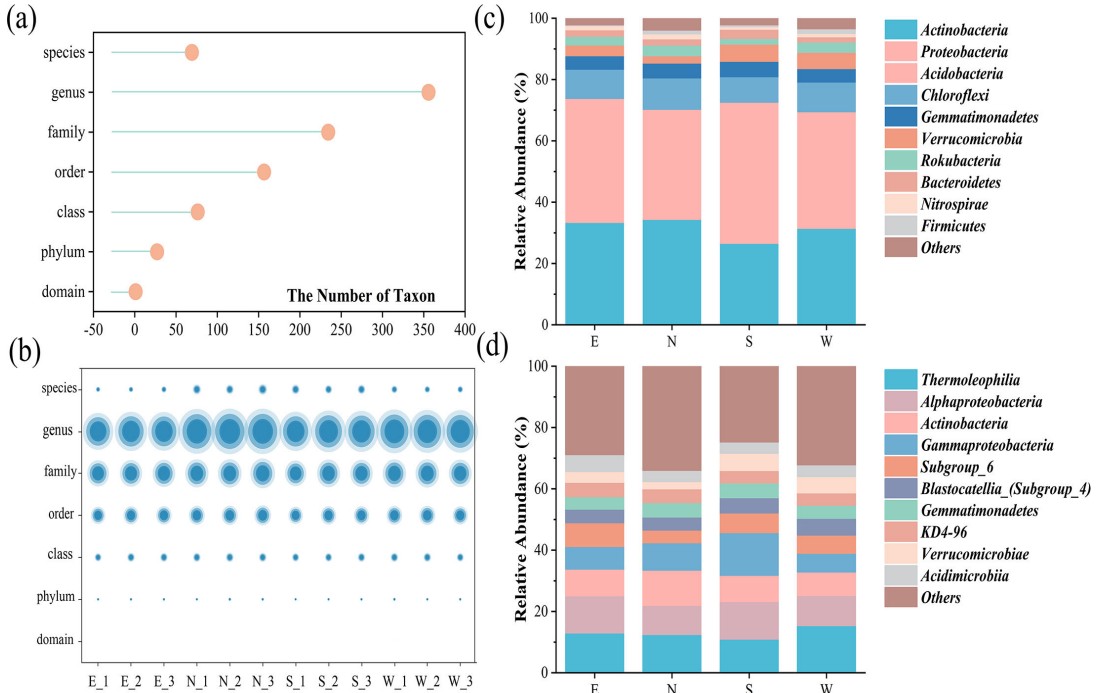

**FIG 2** Taxonomic classifications of bacterial communities in soils. (a, b) Number of species in different levels. (c, d) Microbial community composition at phylum (c) and class (d) levels.

the genus level to assess the differences in community structure. Group south was clustered in different branches with the other three groups. The results indicated that the bacterial microbiota in south soils differed significantly from the other three locations. The soil samples from east and west formed a separate branch (Fig. 3b). Taken together, the samples of the four regions were quite different, echoing the results of PCoA, and had a unique community structure.

To explore the specific differences in soil bacterial communities among different groups, we performed LEfSe analysis to identify abundant bacterial groups in soil samples around uranium mines. Here, p, c, o, f, and g represent phylum, class, order, family, and genus taxonomic levels, respectively. The LDA histogram (LDA score >4) showed microorganisms with significant differences in each group, and species with substantial differences in all soil samples were annotated. Actinobacteria and Proteobacteria were unique biomarkers at the phylum level (Fig. 3c). Twenty genera, including four abundant bacterial clades in east, five in north, nine in south, and two in west soils, were detected (Fig. 3d).

## Correlation between bacterial community and environmental factors

Redundancy analysis (RDA) was performed to reveal the potential association between physicochemical properties and bacterial microbiota of soils. The results of the top 10 colonies with relative abundance at the phylum level showed that all soil samples were well aggregated. Sample group east was positively correlated with uranium and water contents. Sample group north was positively correlated with TP and AP. Sample group west was significantly positively correlated with pH and negatively correlated with EC. MC and pH were positively correlated with uranium. EC and AN were negatively correlated with uranium. UC was significantly positively correlated with the relative abundance of the top 10 bacterial communities of east, north, and west sample groups (Fig. 4a). Analysis of the correlation heatmap between phylum-level bacteria and soil physicochemical properties revealed differences in the sensitivity of bacterial communities to soil physicochemical properties. Uranium content of soils was positively correlated with Firmicutes and Cyanobacteria (Fig. 4b).

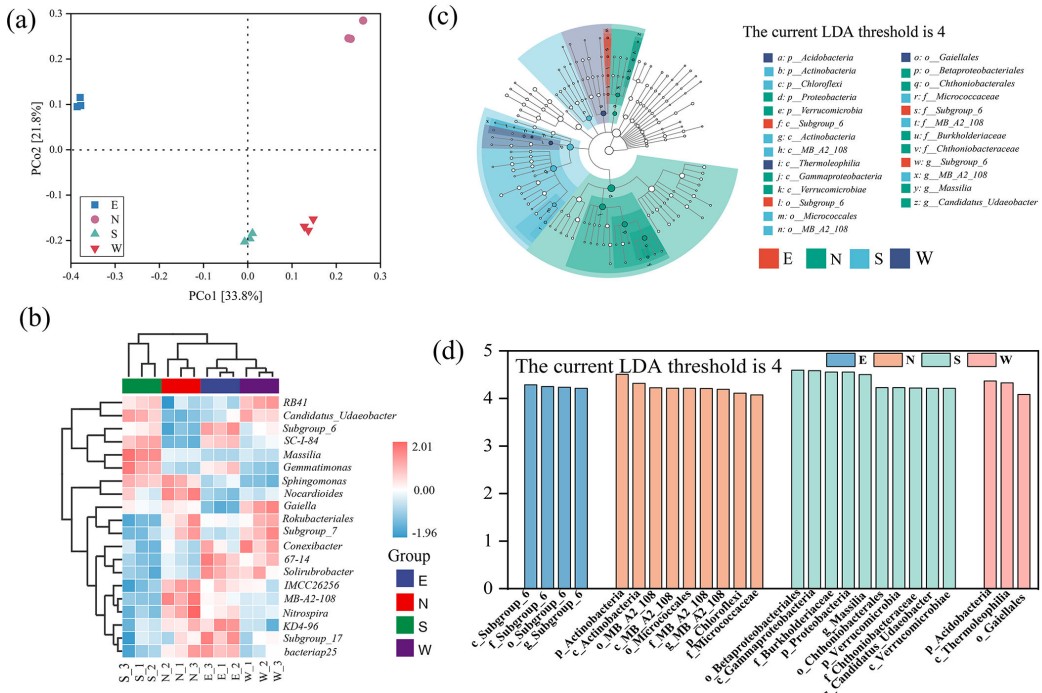

**FIG 3** Analysis of the species differences among soil samples. (a) Principal coordinates analysis of bacterial communities in soils based on the unweighted UniFrac distance. (b) Heatmap analysis of 20 genera with the highest abundance in different samples. (c) Annotated branching map of different species. (d) Linear discriminant analysis score from linear discriminant analysis effect size. Species with significant differences have an LDA score greater than the estimated value, and the default score is 4.0.

## Function prediction analysis for soils microbiota by PICRUSt2

16S rDNA sequencing data obtained from the soil samples were used to predict the metabolic functions of soil bacterial communities by comparing them with a database of microbial genes with known metabolic functions. The results of the PICRUSt2 function prediction were categorized into three levels. The first level had seven main functional categories: biosynthesis, degradation, detoxification, generation of precursor metabolite and energy, glycan pathways, metabolic clusters, and macromolecule modification. Among them, biosynthesis, degradation, and glycan pathways were the most abundant functions (Fig. 5a).

Sixty secondary KEGG Orthology groups were identified in bacterial communities distributed in seven metabolic pathways. The abundance of the predicted gene secondary functional layer was analyzed. Biosynthesis accounted for the largest share at the first level. At the secondary level of biosynthesis, amino acid biosynthesis, cofactor biosynthesis, and fatty acid and lipid biosynthesis (FALB) accounted for the most significant proportion (Fig. 5b). Sample group south differed significantly from the other three groups in amino acid biosynthesis and FALB.

## DISCUSSION

Uranium mining occupies a pivotal role in the progression of national defense and nuclear industries, functioning as a fundamental cornerstone upon which these vital sectors are built. Nevertheless, the extraction process has the potential to result in uranium contamination spreading into the soil, ultimately inflicting widespread and enduring damage upon the flora and fauna in the vicinity (31, 32). Therefore, the remediation of contaminated soil is crucial. Exploring the characteristics of microbial communities in the contaminated area and potential functional microorganisms for uranium tolerance is essential to maximize the advantages of indigenous bacteria for the

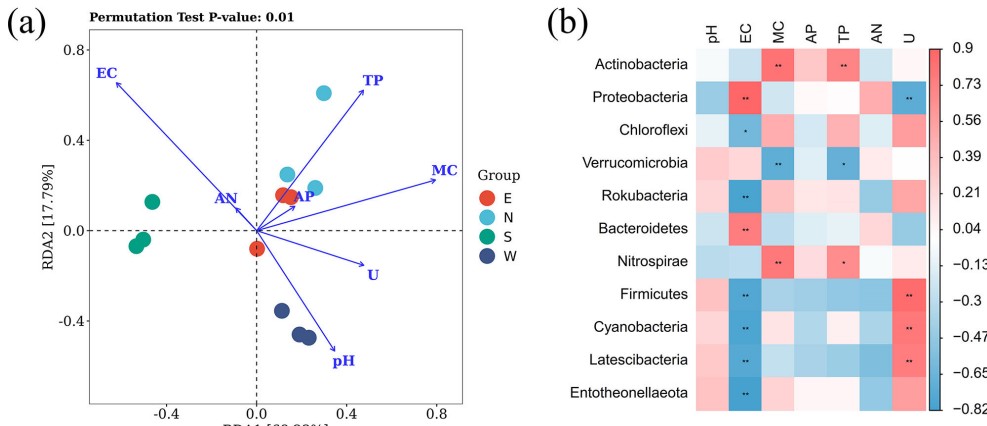

**FIG 4** Correlation analysis between bacterial community and soil environmental factors. (a) Redundancy analysis of correlations between the abundance, richness, and diversity of microbial community and soil environmental variables. (b) Correlations between environmental factors and dominant phyla.

bioremediation of specific uranium mines. The abundance of microorganisms in uranium mine soils is recognized as a primary reservoir of strains capable of bioremediation in uranium-contaminated environments. This study employed soil samples collected from the Zoige uranium mine to uncover the potential functional microbiota by exploring the physicochemical properties, bacterial community of the soil, and their correlations.

## Physicochemical properties influence the bacterial microbiota of uranium-contaminated soil

The average uranium content in these soils was 38.20 µg·g$^{-1}$, representing a remarkable 13.7-fold and 19.1-fold increase compared to the background value of Chinese soil and global soil, respectively (33). This suggests that the soils may be significantly contaminated by uranium. Some studies have shown that soils are affected by atmospheric transport, rainwater leaching transport, groundwater transport, and other factors, and waste uranium from the uranium mining process can diffuse to the surrounding soils, ultimately leading to localized accumulation (34). Thus, UC differs significantly in different surrounding areas. Generally, the higher the uranium contamination in the tailings area, the lower the bacterial diversity (15). Our data showed that the order of bacterial abundance of the four regions was south > east > west > north, with significant differences between south and north regions. According to the present study, the effect of uranium contamination on soil physicochemical and bacterial diversity at different locations did not show a similar trend. This suggests that the relationship between soil physicochemical properties, uranium concentration, and soil microorganisms is not purely linear and is influenced by various factors.

Analysis of species composition showed that shared bacteria were fewer than endemic bacteria among the bacterial communities. This indicates that different soils differ significantly in bacterial communities, consistent with the results of β-diversity. Research shows that the bacterial communities are pivotal participants of the biogeochemical cycles and serve as highly sensitive indicators of environmental changes (35). Bacterial community composition in heavily uranium-contaminated soils is an essential indicator of the health of the environment in uranium mining regions (36). Overall, variation in the bacterial structure in different regions under long-term exposure to high concentrations of uranium offers more possibilities for functional bacterial exploration.

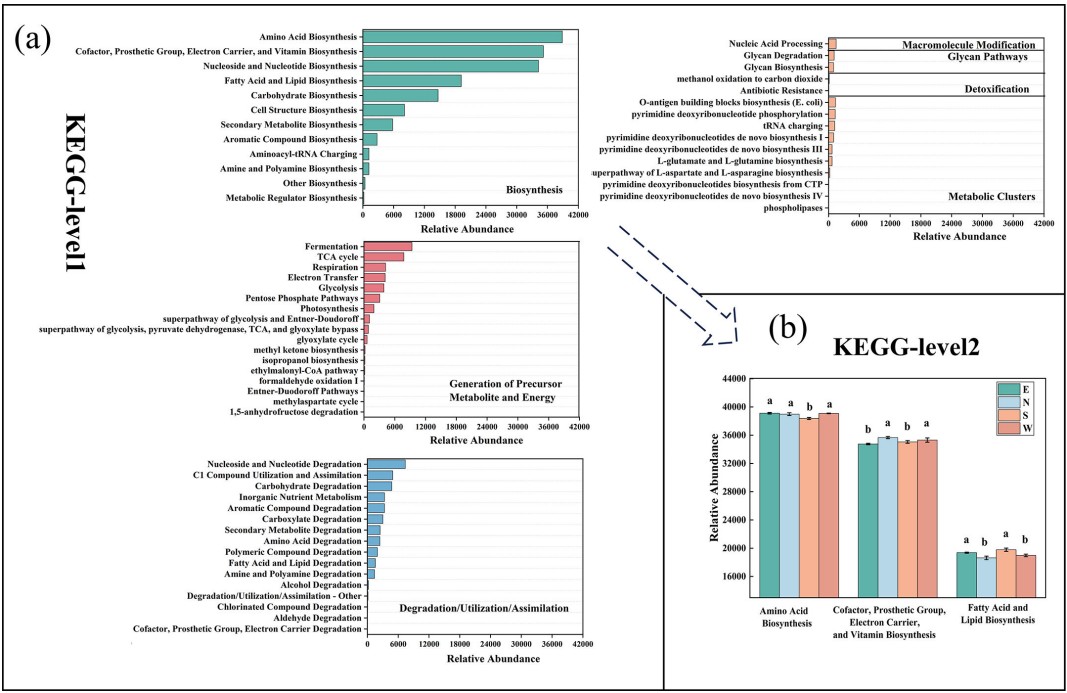

**FIG 5** Function prediction of microbiota by PICRUSt2 analysis. Functional abundances at primary (a) and secondary (b) levels by KEGG annotations.

## Functional microbiota for uranium sequestration and their influencing factors

It is well known that uranium contamination can alter soil microbial communities and their relative abundance. Numerous studies have analyzed the community composition and structure of soil bacteria from different uranium mines, reporting differences in dominant species from one uranium mine to another. For example, Li et al. (36) found that *Thiobacillus*, *Sphingomonas*, and *Sulfuriferula* were the dominant species in uranium soils from the Qinghai-Tibet plateau. Mumtaz et al. (37) noted that Proteobacteria, Actinobacteria, Acidobacteria, and Verrucomicrobia were the most abundant phyla in the Ranger uranium mine in northern Australia. In addition, Acidobacter and Bacteroides were the most numerically abundant phyla in the Edgemont and North Cave Hills mines (10). The current study found that Actinobacteria, Proteobacteria, and Acidobacteria were the most abundant phyla in the Zoige uranium mine. Numerous studies have shown that these bacteria are predominant in uranium-rich soils. In addition, our data showed that each of the four regions had its signature species. Further analysis showed that Actinobacteria, Proteobacteria, and Acidobacteria exhibited significant variation. Actinobacteria can reduce uranium migration and provide bioactive substances by feeding on certain microorganisms (38). Acidobacteria can enhance the metal fixation capacity of soil mainly by regulating soil physicochemical properties (39). Proteobacteria play a major role in the remediation of uranium-contaminated soils. They have been utilized in uranium remediation of uranium-contaminated soils in composting processes (40, 41). Therefore, the composition of bacterial communities in long-term heavy metal-contaminated soils is jointly influenced by the type of land use and soil physicochemical properties. These bacteria evolve into communities that are stable and functionally tolerant to uranium (37).

The physicochemical properties of soil influence the composition and dynamic succession of microbiota. Generally, soil nutrients, including ammonium nitrogen, total nitrogen, and total phosphorus, are crucial for bacterial growth and reproduction, serving as pivotal factors that influence microbial community structure (42). The present study found that uranium content was positively correlated with pH, water content,

and TP. However, no direct correlation was found between the distribution of bacterial communities and soil physicochemical properties in soils exposed to high concentrations of uranium. It was found that differences in the composition of bacterial communities between different soil habitats are determined by their complex environmental factors (43). Therefore, further studies should explore the interplay between dominant soil bacterial strains and uranium concentration.

Furthermore, our results have unveiled a significant positive correlation between the UC and the abundances of Firmicutes and Cyanobacteria in Zoige uranium mine soils. It has been shown that Cyanobacteria can regenerate naturally after extensive cell lysis and uranium biomineralization under sustained uranium exposure and oxygenated phosphate limiting conditions (44). In addition, Firmicutes, capable of swiftly responding to heavy metal stress and rapidly adapting to extreme environments, often dominate uranium mining soils and play a pivotal role in soil remediation (45). For example, the introduction of *Bacillus globulus*, a species belonging to the Firmicutes phylum, significantly altered the overall bacterial structure of the soil, particularly by increasing the abundance of Firmicutes, ultimately achieving more effective soil remediation (12). Therefore, they can easily survive and propagate in the tailings and remediation areas with severe heavy metal contamination and poor soil nutrient content compared with other bacteria (46). Further studies should screen more strains and explore their interactions and mechanisms underlying their tolerance in uranium-contaminated environments to provide more insights into developing efficient and inexpensive remediation technologies for uranium-contaminated soils.

## Functional responses of bacterial microbiota to uranium-contaminated soils

Functional predictions showed that biosynthesis accounted for the highest proportion at the primary level. In particular, amino acids biosynthesis and fatty acids biosynthesis accounted for the highest proportion at the secondary level (Fig. 5b). It has been shown that phospholipids, lipopeptides, glycolipids, fatty acids, and lipoproteins, as well as particles and polymeric organisms, form metal complexes at the soil-water interface. They are desorbed by lowering the tension at the soil-water interface (47, 48). Thus, macromolecular synthesis associated with enrichment would be more abundant in soil bacterial communities under high levels of uranium contamination. So the biosynthesis processes may be associated with the uranium tolerance capabilities of bacterial communities, potentially augmenting uranium sequestration through the enhancement of metabolic functions and the utilization of amino acids, fatty acids, reducing enzymes, and cofactors. This not only reduces the migration of uranium in the soil and minimizes ecosystem damage but is also critical for bacterial growth and development in highly enriched uranium environments (21, 49, 50).

Since uranium is the primary heavy metal in uranium mine soil, the microbial community exhibits a series of physiological responses to cope with uranium stress and form specific microbial flora (42). Therefore, analyzing the functional profiles of these microbial flora may help us gain a deeper understanding of the potential ecological impacts under high-level uranium contamination. In the present study, the dominant bacteria in Zoige uranium mine soil were mainly chemoenergetic heterotrophs, which play specific driving roles and ecological functions in the ecosystem. For instance, bacteria can sequestrate uranium through chelation or complexation, thereby reducing the absorption and utilization of uranium in soil by organisms (14). Unfortunately, in practical large-scale applications, these sequestration processes are influenced by various factors, particularly the interactions among microbiota in the soil remediation system (6, 51). Therefore, studying the functions of the entire bacterial community is beneficial for subsequent parameter adjustment and functional optimization required for microbial remediation.

## Conclusion

In summary, high-throughput sequencing and bioinformatic analysis revealed functional microbiota for uranium sequestration in Zoige uranium mine soils. Actinobacteria, Firmicutes, and Cyanobacteria were the potential functional microbiota in Zoige uranium mine soils. These microbes interacted and tolerated uranium via amino acid and fatty acid biosynthesis pathways. These findings provide insights into the functional microbiota of uranium sequestration, which are conducive to developing microbial resources and bioremediation technology for treating uranium contamination. Nevertheless, further research should be devoted to demonstrating the correlations between functional microbiota and uranium interactions using meta-omics approaches.

### ACKNOWLEDGMENTS

This work was funded by the Doctoral Scientific Fund Project of Southwest University of Science and Technology (20zx7130) and the Open Fund of Fundamental Science on Nuclear Wastes and Environmental Safety Laboratory (22kfhk04).

### AUTHOR AFFILIATIONS

[1]College of Life Science and Engineering, Southwest University of Science and Technology, Mianyang, Sichuan, China
[2]Sichuan Institute of Nuclear Geological Survey, Chengdu, China
[3]College of Biomass Science and Engineering, Sichuan University, Chengdu, China
[4]Fundamental Science on Nuclear Wastes and Environmental Safety Laboratory, Southwest University of Science and Technology, Mianyang, Sichuan, China

### AUTHOR ORCIDs

Guiqiang He http://orcid.org/0000-0002-1571-0465

### FUNDING

| Funder | Grant(s) | Author(s) |
| --- | --- | --- |
| Doctoral Scientific Fund Project of Southwest University of Science and Technology | 20zx7130 | Guiqiang He |
| Open Fund of Fundamental Science on Nuclear Wastes and Environmental Safety Laboratory | 22kfhk04 | Xu Zhang |

### AUTHOR CONTRIBUTIONS

Xiang Wang, Data curation, Formal analysis, Investigation, Methodology, Visualization, Writing – original draft | Li Zhao, Resources | Xu Zhang, Supervision, Validation | Yanxia Wei, Data curation, Methodology | Aixia Lu, Investigation, Methodology, Project administration | Jian Zhou, Methodology, Writing – review and editing | Guiqiang He, Conceptualization, Funding acquisition, Supervision, Writing – review and editing

### DATA AVAILABILITY

Raw sequence data are available on NCBI Sequence Read Archive (BioProject accession no. PRJNA1101590). Sequence Read Archive (SRA) data can be accessed via https://www.ncbi.nlm.nih.gov/bioproject/PRJNA1101590/.

### ADDITIONAL FILES

The following material is available online.

#### Supplemental Material

**Supplemental material (Spectrum02517-24-S0001.docx).** Fig. S1 and S2.

Open Peer Review

PEER REVIEW HISTORY (review-history.pdf). An accounting of the reviewer comments and feedback.

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
