## [Reviewer comments · Microbiology Spectrum]

Microbiology Spectrum

Exploring functional microbiota for uranium sequestration in Zoige uranium mine soil

Xiang Wang, Li Zhao, Xu Zhang, Yanxia Wei, Aixia Lu, Jian Zhou, and Guiqiang He

Corresponding Author(s): Guiqiang He, Southwest University of Science and Technology School of Life Sciences and Engineering

Review Timeline:

Submission Date:	October 6, 2024
Editorial Decision:	November 13, 2024
Revision Received:	January 9, 2025
Editorial Decision:	January 20, 2025
Revision Received:	February 26, 2025
Accepted:	March 16, 2025

Editor: Philips Akinwale

Reviewer(s): Disclosure of reviewer identity is with reference to reviewer comments included in decision letter(s). The following individuals involved in review of your submission have agreed to reveal their identity: Xugang Dang (Reviewer #1)

Transaction Report:

DOI: <https://doi.org/10.1128/spectrum.02517-24>

Re: Spectrum02517-24 (Exploring functional microbiota for uranium sequestration in Zoige uranium mine soil)

Dear Dr. Guiqiang He:

Thank you for the privilege of reviewing your work. Below you will find my comments, instructions from the Spectrum editorial office, and the reviewer comments.

Revision Guidelines

Sincerely,
Philips Akinwale
Editor
Microbiology Spectrum

Reviewer #1 (Comments for the Author):

In this manuscript, functional microbiota for uranium sequestration in uranium mine soil were revealed. The results are important to help develop in situ remediation technology for uranium contamination. The design of the experiment is strongly correlated with the field of microbiology, the results are depicted and discussed well.

1. There are some errors in citations and references. Lines 338-340, 351, 369, 420, 502, and 513. Please check throughout the

full text.

2. The citation and description of figure 7c is missing.
3. The spelling of Latin name for microbes is confusing. Please correct with Phylum orthodox and Genus italic.
4. Other minor issues for format, lines 31, 33, 49, 57, 63, 66, 71, 421. Please check.
5. Line 402, correct the word "uranium enrichment" gives another meaning. Accumulation or sequestration can be used.
6. Lines 414-416, the statement "Numerous studies have shown that interactions between bacteria and uranium lead to chelation or complexation, which reduces the concentration of toxic metals in the environment". The concentration of toxic metals is steady in the environment. The authors need to be rephrased to convey the intended meaning accurately.
7. Although the revelation of functional microbiota for uranium sequestration is important for environment remediation, the functional strains and interaction mechanism with uranium should be explored in future studies.

Reviewer #2 (Comments for the Author):

This manuscript titled "Exploring functional microbiota for uranium sequestration in Zoige uranium mine soil" explored the potential functional microbiota for uranium sequestration in Zoige uranium mine soil by using high-throughput sequencing and bioinformatics analysis. This is a topic of interest to researchers in related fields, but this research is not written well and the innovation is insufficient. The paper needs revision and my detailed comments are as follows:

1. The grammar of the manuscript needs to be carefully checked and revised.
2. The abstract should be rewrite to highlight the novelty of this manuscript.
3. The logic of the introduction needs to be improved.
4. The content of Figure 1 is too simple and should be presented in the supplementary materials.
5. Figure 2 should be removed and the data in Figure 2 should be presented in a table.
6. Figure 3 (b) cannot be found in the title.
7. In the "Material and Methods" section. The paper starts from section 2.2 The "Determination of soil physical properties" section jumps directly to 2.4 The "DNA extraction, PCR amplification, and amplitude detection" section, Section 2.3 is missing.
8. In the "Results" section. In "3.1. Determination of soil physicochemical properties", the paper mentions that there are significant differences in TP, AP, and UC in soils in the four regions, and it is recommended that specific significance levels should be provided.
9. In "3.3. Soil microbial community structure", "Actinobacteria" was expressed as "Actinobaderia", a correction should be proved.
10. Please unify the full name of snail throughout the article.

In this manuscript, functional microbiota for uranium sequestration in uranium mine soil were revealed. The results are important to help develop in situ remediation technology for uranium contamination. The design of the experiment is strongly correlated with the field of microbiology, the results are depicted and discussed well.

1. There are some errors in citations and references. Lines 338-340, 351, 369, 420, 502, and 513. Please check throughout the full text.
2. The citation and description of figure 7c is missing.
3. The spelling of Latin name for microbes is confusing. Please correct with Phylum orthodox and Genus italic.
4. Other minor issues for format, lines 31, 33, 49, 57, 63, 66, 71, 421. Please check.
5. Line 402, correct the word “uranium enrichment” gives another meaning. Accumulation or sequestration can be used.
6. Lines 414-416, the statement “Numerous studies have shown that interactions between bacteria and uranium lead to chelation or complexation, which reduces the concentration of toxic metals in the environment”. The concentration of toxic metals is steady in the environment. The authors need to be rephrased to convey the intended meaning accurately.
7. Although the revelation of functional microbiota for uranium sequestration is important for environment remediation, the functional strains and interaction mechanism with uranium should be explored in future studies.

Comments:

This manuscript titled “Exploring functional microbiota for uranium sequestration in Zoige uranium mine soil” explored the potential functional microbiota for uranium sequestration in Zoige uranium mine soil by using high-throughput sequencing and bioinformatics analysis. This is a topic of interest to researchers in related fields, but this research is not written well and the innovation is insufficient. The paper needs revision and my detailed comments are as follows:

1. The grammar of the manuscript needs to be carefully checked and revised.
2. The abstract should be rewrite to highlight the novelty of this manuscript.
3. The logic of the introduction needs to be improved.
4. The content of Figure 1 is too simple and should be presented in the supplementary materials.
5. Figure 2 should be removed and the data in Figure 2 should be presented in a table.
6. Figure 3 (b) cannot be found in the title.
7. In the "Material and Methods" section. The paper starts from section 2.2 The "Determination of soil physical properties" section jumps directly to 2.4 The "DNA extraction, PCR amplification, and amplitude detection" section, Section 2.3 is

missing.

8. In the "Results" section. In "3.1. Determination of soil physicochemical properties", the paper mentions that there are significant differences in TP, AP, and UC in soils in the four regions, and it is recommended that specific significance levels should be provided.
9. In "3.3. Soil microbial community structure", "*Actinobacteria*" was expressed as "*Actinobaderia*", a correction should be proved.
10. Please unify the full name of snail throughout the article.

Reviewer #1:

In this manuscript, functional microbiota for uranium sequestration in uranium mine soil were revealed. The results are important to help develop in situ remediation technology for uranium contamination. The design of the experiment is strongly correlated with the field of microbiology; the results are depicted and discussed well.

Response: Thanks very much for the approval of our research work. We also appreciate the time and effort you have dedicated to providing constructive comments to improve our paper. We have carefully modified the manuscript according to your suggestions for improving the quality of manuscript. All the revisions were **marked in red** in the revised manuscript.

1. There are some errors in citations and references. Lines 338-340, 351, 369, 420, 502, and 513. Please check throughout the full text.

Response: Thank you very much for mentioning this issue. We have attentively revised the formats of citations and references in the revised manuscript following the standard form provided by this journal. Please see the lines 338-340, 352, 368, 417, 437-632 in the revised manuscript.

2. The citation and description of figure 7c is missing.

Response: We are very sorry to omit citation and description of Fig. 7c in the manuscript. Firstly, we have changed the figure number according to the suggestions by another reviewer, and the Fig. 7c has been modified to Fig. 5c. In addition, we have

modified and added the related sentences to describe the Fig. 5c in the revised manuscript. Please see the lines 404-405 in the revised manuscript.

3. The spelling of Latin name for microbes is confusing. Please correct with Phylum orthodox and Genus italic.

Response: Thank you very much for your attention to this issue. We have carefully corrected the relevant spellings of Latin name for microbes in the full text. For example, please see the lines 33-34, 36, 43, 76-79, 85, 225-232, 255-256, 273, 339-345, and 348-352 in the revised manuscript.

4. Other minor issues for format, lines 31, 33, 49, 57, 63, 66, 71, 421. Please check.

Response: Thanks very much for your kind work and useful comments to our manuscript. We have earnestly checked and modified the relevant sentences according to your suggestions. Please see the lines 33-34, 64, 71, 417 in the revised manuscript.

5. Line 402, correct the word "uranium enrichment" gives another meaning. Accumulation or sequestration can be used.

Response: We are appreciate for your useful comments to our work. We have replaced the word and rewritten the sentences according to your suggestion. Please see the lines 397-401 in the revised manuscript.

6. Lines 414-416, the statement "Numerous studies have shown that interactions

between bacteria and uranium lead to chelation or complexation, which reduces the concentration of toxic metals in the environment". The concentration of toxic metals is steady in the environment. The authors need to be rephrased to convey the intended meaning accurately.

Response: We are very appreciated for your efforts and useful comments to our manuscript. We have modified the related sentences according to your suggestion. Please see the lines 413-415 in the revised manuscript.

7. Although the revelation of functional microbiota for uranium sequestration is important for environment remediation, the functional strains and interaction mechanism with uranium should be explored in future studies.

Response: Thanks for your constructive suggestions. We totally agree with your point. In this manuscript, functional microbiota for uranium sequestration was unveiled, which laid a foundation for isolation and identification of functional strains. To better obtain functional strains and develop the bioremediation technology for contamination treatment by uranium, further research should be devoted to revealing the interaction mechanism between the functional strains and uranium. We will explore this field in much greater detail in our future article.

Reviewer #2:

This manuscript titled "Exploring functional microbiota for I uranium sequestration in Zoige uranium mine soil" explored the potential functional microbiota for uranium

sequestration in Zoige uranium mine soil by using high-throughput sequencing and bioinformatics analysis. This is a topic of interest to researchers in related fields, but this research is not written well and the innovation is insufficient. The paper needs revision and my detailed comments are as follows:

Response: Thank you very much. We want to extend our appreciation for taking the time and energy to provide such insightful guidance on our manuscript. We have carefully modified the manuscript according to your suggestions for improving the quality of manuscript. All the revisions were **marked in red** in the revised manuscript.

1. The grammar of the manuscript needs to be carefully checked and revised.

Response: Thank you for providing the useful comments to our manuscript. Firstly, this manuscript had been recently polished, as shown in Fig. 1. In addition, we have invited two native speakers of English to revise the terminology throughout the text as appropriate. At last, we have revised the manuscript carefully and tried to avoid any grammar or syntax error.

Fig. 1 Editorial certificate of article polishing.

2. The abstract should be rewritten to highlight the novelty of this manuscript.

Response: We are very much appreciate your efforts and useful comments to our manuscript. To better highlight the novelty of this manuscript, the “**Abstract**” section has been modified according to your suggestions. Please see the lines 24-46 in the revised manuscript.

3. The logic of the introduction needs to be improved.

Response: Thank you for providing the useful comments to our manuscript. Based on your insightful feedback, we have refined the logic of “**Introduction**” in the revised manuscript. Based on the significance of the Zoige uranium mine and its unique ecological environment, this study emphasizes the necessity of *in-situ* bioremediation. Consequently, our research endeavors to unravel the functional microbiota involved in uranium interactions and their roles in enrichment sequestration within these soils. Please see the lines 62-102 of the “**Introduction**” section in the revised manuscript.

4. The content of Figure 1 is too simple and should be presented in the supplementary materials.

Response: Thanks very much for your useful comments on our work. This figure has been exhibited in the “**Supplementary materials**” as Fig. S1 according to your suggestions.

5. Figure 2 should be removed and the data in Figure 2 should be presented in a table.

Response: Thanks very much for your kind work. The data of soil physicochemical properties in Figure 2 have been exhibited in **Table 1**. The corresponding descriptions as shown the lines 182-193 in the revised manuscript.

Table 1 Physicochemical properties of soil samples (mean \pm standard error).

Physicochemical properties	East	North	South	West
pH	6.71 \pm 0.22a	6.69 \pm 0.18a	6.67 \pm 0.06a	6.90 \pm 0.01a
EC (μ s/cm)	145.60 \pm 14.77ab	126.75 \pm 7.83b	165.1 \pm 19.74a	44.05 \pm 3.40c
MC (%)	22.26 \pm 0.31a	15.60 \pm 0.11b	4.41 \pm 0.18d	12.71 \pm 0.15c
TP (mg/100mg)	16.88 \pm 0.02a	14.50 \pm 0.01b	10.75 \pm 0.01c	10.25 \pm 0.04d
AP (mg/100mg)	0.37 \pm 0.04a	0.16 \pm 0.01b	0.16 \pm 0.01b	0.16 \pm 0.02b
AN (mg/100mg)	5.07 \pm 0.06a	4.24 \pm 0.01b	4.03 \pm 0.02d	4.15 \pm 0.01c
UC (mg/g)	35.20 \pm 0.08d	40.31 \pm 0.02b	36.73 \pm 0.03c	40.62 \pm 0.08a

6. Figure 3 (b) cannot be found in the title.

Response: We are very sorry for omitting the Fig. 3(b) in the title. We have added the annotation in the revised manuscript, and please see the lines 247 and 637. Corresponding results were described in lines 244-248.

7. In the "Material and Methods" section. The paper starts from section 2.2 The "Determination of soil physical properties" section jumps directly to 2.4 The "DNA extraction, PCR amplification, and amplitude detection" section, Section 2.3 is missing.

Response: We are very sorry for our incorrect writing. Thank you very much for your attention to this issue. We have revised the related sentences, and please see the lines 133, 144, 158, and 173 in the revised manuscript.

8. In the "Results" section. In "3.1. Determination of soil physicochemical properties", the paper mentions that there are significant differences in TP, AP, and UC in soils in the four regions, and it is recommended that specific significance levels should be provided.

Response: Thank you for providing the positive suggestions to our study. In this study, a one-way analysis of variance (ANOVA) was carried out to evaluate significant differences ($P < 0.05$) in physicochemical properties between the soil samples using SPSS 19.0 software. The different letters indicated significant differences among the samples in the **Table 1**. Please see the lines 186-191 in the revised manuscript.

9. In "3.3. Soil microbial community structure", "Actinobacteria" was expressed as "Actinobaderia", a correction should be proved.

Response: Thanks very much for your kind work. We have modified the word in the revised manuscript, and please see the lines 227 in the revised manuscript.

10. Please unify the full name of samples throughout the article.

Response: We appreciate the time and effort you have dedicated to providing useful comments to improve our paper. We are very sorry to confuse you by not unify the full name of samples throughout the article. We standardized the full name of the sample, and please see the relevant sections marked in red in the revised manuscript.

Re: Spectrum02517-24R1 (Exploring functional microbiota for uranium sequestration in Zoige uranium mine soil)

Dear Dr. Guiqiang He:

Thank you for the privilege of reviewing your work. Below you will find my comments, instructions from the Spectrum editorial office, and the reviewer comments.

Thank you for submitting your revised manuscript. Following your thorough revisions in response to the reviewers' comments, an additional review is necessary to further enhance the quality of your manuscript. Attached are the suggestions and comments for modification. Thank you for your continued patience throughout this process.

Revision Guidelines

Sincerely,
Philips Akinwale
Editor
Microbiology Spectrum

Spectrum02517-24R1

1. Line 73: What is “it” referring to? As it is not clear from the prior sentence.

Response: Thank you for highlighting the clarity issue with the pronoun “it” on Line 73. To address your concern, “it” in this context is referring to the research cited in the references. We apologize for any confusion this may have caused and we have revised the sentence to ensure clarity. Please see the lines 80-83 in the revised manuscript.

2. Line 84: *Delightedly* will be a more appropriate word here.

Response: We appreciate your careful review of my manuscript and the valuable feedback you have provided. Regarding your suggestion, we fully concur. Your insight has greatly improved the clarity and expressiveness of the sentence. I have already replaced the word according to your recommendation. Please see the line 91 in the revised manuscript.

3. Line 85: In “For instances, such as *Bacillus, Actinomycetes*” **such as** should be deleted.

Response: We are grateful for your help in improving the clarity and accuracy of the paper. Regarding your suggestion to delete “such as” on Line 85, we fully understand and agree with your point. The sentence can indeed be simplified without losing any meaning. I have made the necessary correction in the manuscript accordingly. Please see the line 92 in the revised manuscript.

4. **Introduction:** to broaden readability of this article, authors should include various physical, chemical, and biological methods that have been applied to remediate environmental uranium pollution. Why is this study essential in view of previous applications and techniques?

Response: In response to your proposal to enhance the readability of article by incorporating a range of physical, chemical, and biological methods utilized for uranium contamination remediation, we entirely agree. We have added and modified the appropriate sentences according to your suggestions. Please see the lines 71-80 and 94-98 in the revised manuscript. Below is the reason why this study remains indispensable within the backdrop of previous applications and techniques: In terms of uranium-contaminated soil remediation, although physical and chemical remediation technologies boast high removal efficiency and operational simplicity, they are often accompanied by relatively high costs and a tendency to induce secondary pollution (Lines 76-78). Bioremediation possesses advantages including safety and cost-effectiveness, earning it the reputation of being an environmentally friendly alternative technology (Lines 79-80). But considering the adaptability of strains during the remediation process, bioremediation employing indigenous strains may achieve superior outcomes. So it is crucial to uncover the microbial community and pinpoint the functional microbiota in the contaminated soil (Lines 94-98). Thank you for your constructive feedback, which has greatly enhanced the focus and relevance of our work.

5. Table 1: and lines 184 - 185 need to be consistent with other parameters listed, organic matter should be abbreviated as OM (and not MC which is designated as moisture content in the Materials and Methods section). OM or MC should be clarified.

Response: Thank you for bringing this inconsistency to our attention. We apologize for the confusion caused by the abbreviation misuse. In this manuscript, abbreviation “MC” stands for “Moisture Content”, and we have modified the entire manuscript. Please see the lines 131, 192,

194, and Table 1 in the revised manuscript. We appreciate your careful review and valuable feedback, which significantly improve the accuracy and readability of our work.

6. Line 134: the information on the DNA kit manufacturer should be included.

Response: Thank you for your valuable suggestion. We have now included the manufacturer information for the DNA kit, and please see the lines 141-142 in the revised manuscript. This addition aims to provide a more comprehensive understanding of the materials and methods used in our study. We appreciate your careful review and consideration.

7. Line 206: "...south, and had 7678, 6542, 6110, and 6997 ASVs, respectively" the first and should be removed.

Response: Thank you for pointing out the redundancy in line 206. We have revised the sentence by removing the first "and" to ensure clarity and conciseness. Please see the lines 213-214 in the revised manuscript. We appreciate your careful review and valuable feedback.

8. Lines 237-238: "In the PCoA plot, PC1 and PC2 explained 59.2% and 12.6% of the variance, respectively (Fig. 3a)." However, the % value indicated on this graph (Fig. 3a) shows that PCo1 and PCo2 explained 33.8% and 21.8% of the variance, respectively. Authors should clarify this discrepancy.

Response: Thank you for bringing this discrepancy to our attention. Upon careful examination, we acknowledge that there is a mismatch between the text description and the information presented in Fig. 3a. We apologize for this error, which may have arisen due to a mix-up during the preparation of the manuscript. We have updated the text description to accurately reflect the

percentage values shown in Fig. 3a, and please see the lines 245-247 in the revised manuscript. We have also double-checked all other instances of numerical data in the manuscript to ensure their accuracy and consistency with the figures and tables presented. We apologize for any confusion this may have caused and appreciate your careful review.

9. Lines 239-240: The statement implies that non-overlapping ASV distributions and dissimilar community structures automatically indicate reliability. Authors should revise this conclusion. The reliability of results depends on factors such as proper experimental design, replication, and control of biases, not merely on the observed dissimilarity. Reliability typically refers to the reproducibility and consistency of results across multiple experiments or sampling efforts. The statement conflates ecological patterns with methodological rigor. Authors should consider statement such as - "The non-overlapping ASV distributions and dissimilar community structures observed among the four sampling points suggest distinct microbial communities"

Response: Thank you for bringing this concern to our attention. Upon reflection, we acknowledge that our previous statement may have implied an overly simplistic connection between non-overlapping ASV distributions, dissimilar community structures, and the reliability of our results. Reliability, as you correctly point out, typically refers to the reproducibility and consistency of results across different experiments or sampling efforts. We agree that our statement conflated ecological patterns with methodological rigor, which is not an accurate reflection of our research approach. We have modified the sentences according to your suggestions, and please see the lines 247-249 in the revised manuscript. We thank you again for your valuable insights and appreciate your careful review of our manuscript.

10. Fig. 5c was not interpreted or referred to in the Results section and therefore should be removed or should be interpreted if needed to be included in the manuscript. Moreover, it was vaguely discussed in the Discussion section.

Response: Thank you for bringing this to our attention. Upon reviewing our manuscript, we acknowledge that Fig. 5c was indeed not adequately referred to in the Results section. We apologize for any confusion this may have caused. We have removed Fig. 5c from the revised manuscript according to your suggestion. Additionally, we have revised the Discussion section to provide a more detailed and nuanced discussion of figures and data presented in our work. Please see the lines 413-426 and red-marked sections of Discussion in the revised manuscript.

11. Line 296: at least a citation should be included here.

Response: Thank you for your valuable feedback on our manuscript. In response to your suggestion on Line 296, we acknowledge that a citation would indeed enhance the rigor and credibility of the statement. We have now modified the sentences and included relevant citations to support the assertion made in that line. Please see the lines 302-306 in the revised manuscript. We appreciate your guidance in improving the quality of our work.

12. Line 309: "...UC in these soils was 38.20 mg·g⁻¹, which is an exceptionally highly ..." a comparison to global/regional uranium concentrations in soil will place 38.20 mg·g⁻¹ in perspective.

Response: Thank you for your insightful comment on Line 309, which has prompted us to provide a more comprehensive context for the uranium concentration value reported. Additionally, we would like to correct a unit error in the text: the uranium concentration should

be expressed as $\mu\text{g}\cdot\text{g}^{-1}$, not $\text{mg}\cdot\text{g}^{-1}$. To better situate the uranium content of $38.20 \mu\text{g}\cdot\text{g}^{-1}$ in perspective, we have included a comparison with the background value of Chinese soil ($2.79 \mu\text{g}\cdot\text{g}^{-1}$) and global soil ($2.00 \mu\text{g}\cdot\text{g}^{-1}$). Please see the lines 319-322 in the revised manuscript. We apologize for the unit error and appreciate your suggestion as it has significantly enriched the interpretation of our results.

13. Lines 372-374 and 413-415: at least a citation should be included here for these studies.

Response: In response to your suggestion, a citation should indeed be included for the studies mentioned in lines 372-374 and 413-415. We apologize for the oversight. We have revised the manuscript to incorporate appropriate references for these studies to ensure that all claims are supported by credible sources and to maintain the rigor and transparency of the work. Please see the lines 383-391 and 413-426 in the revised manuscript. Thank you for bringing this to my attention.

14. Overall, are there specific species or uranium tolerant strains that can be recommended for uranium remediation? see species in Fig. 2a.

Response: Thank you for your inquiry. In this manuscript, our analysis of microbial community primarily focused on the phylum and genus levels. Among the top 10 most abundant species identified, there were several functional species including *Nocardioides halotolerans*, *Massilia eurypsychrophila*, and *Umezawaea tangerina*. Due to the unique climatic conditions of the Zoige uranium mine, these identified species have the potential to be novel, and current research on the interactions between these species and uranium is relatively scarce. However, these species have been demonstrated potential application value in the fields such as bioremediation and

environmental governance (Ma, Y.; Wang, J.; Liu, Y.; Wang, X.; Zhang, B.; Zhang, W.; Chen, T.; Liu, G.; Xue, L.; Cui, X. *Nocardioides*: “Specialists” for Hard-to-Degrade Pollutants in the Environment. *Molecules* 2023, 28, 7433. <https://doi.org/10.3390/molecules28217433>. Yang, E.; Cui, D.; Wang, W. Research progress on the genus *Massilia*. *Microbiology China*, 2019, 46(6): 1537-1548. DOI: [10.13344/j.microbiol.china.180573](https://doi.org/10.13344/j.microbiol.china.180573)). Additionally, *N. halotolerans* belongs to the Actinobacteria phylum, which was identified as a biomarker in this study (Please see the lines 34-35 in the revised manuscript). Both *M. eurypsychrophila* and *U. tangerina* belong to the Proteobacteria phylum, which was the dominant phylum in our analysis (Please see the lines 32-33 in the revised manuscript). In summary, the above three species can be recommended for attempts at uranium contamination remediation. However, it is worth noting that the effectiveness of these species may vary depending on soil conditions, uranium concentration, and other environmental factors. Therefore, it is recommended to conduct site-specific evaluations and trials to determine the most suitable species for a given remediation project. We appreciate your guidance in improving the clarity and completeness of our work.

Re: Spectrum02517-24R2 (Exploring functional microbiota for uranium sequestration in Zoige uranium mine soil)

Dear Dr. Guiqiang He:

Your manuscript has been accepted, and I am forwarding it to the ASM production staff for publication. Your paper will first be checked to make sure all elements meet the technical requirements. ASM staff will contact you if anything needs to be revised before copyediting and production can begin. Otherwise, you will be notified when your proofs are ready to be viewed.

Sincerely,
Philips Akinwale
Editor
Microbiology Spectrum